# An Overview of the Evolution of Capsule Endoscopy Research—Text-Mining Analysis and Publication Trends

**DOI:** 10.3390/diagnostics12092238

**Published:** 2022-09-16

**Authors:** Rebekka Steinmann, Pablo Cortegoso Valdivia, Tanja Nowak, Anastasios Koulaouzidis

**Affiliations:** 1Service Centre InterNationalTransfer, University of Würzburg, 97070 Würzburg, Germany; 2Gastroenterology and Endoscopy Unit, University Hospital of Parma, University of Parma, 43126 Parma, Italy; 3Medical Affairs, 22765 Hamburg, Germany; 4Department of Clinical Research, University of Southern Denmark, 5230 Odense, Denmark; 5Department of Medicine, OUH Svendborg Sygehus, 5700 Svendborg, Denmark; 6Surgical Research Unit, OUH, 5000 Odense, Denmark; 7Department of Social Medicine and Public Health, Pomeranian Medical University, 70-204 Szczecin, Poland

**Keywords:** capsule endoscopy, research, meta-view

## Abstract

There has been a steady increase (annual percentage growth rate of 19.2%, average of 18.3 citations per document) in capsule endoscopy (CE) publications from a global, interdisciplinary research community on a growing range of CE applications over the last 20+ years. We here present the status of CE as a field of research, tracing its evolution over time and providing insight into its potential for diagnostics, prevention and treatment of gastrointestinal (GI) tract diseases. To portray the development of the CE research landscape in the 2000–2021 time span, we analyzed 5764 scientific publications. Analyses were performed using the R language and environment for statistical computing and graphics and VOSviewer, a software developed for scientific literature analysis by scientometricians. The aim of this paper is to provide a wide comprehensive analysis of the trends in CE publications. We thus performed subgroup analysis on the selected papers, including indications, annual percentage growth rate, average citations per document, most publications from research areas/interdisciplinary field of the articles, geography, collaboration networks through institutions, specific clinical keywords and device type. The firm increase in CE publications over the last two decades highlights the overall strength of the technology in GI applications. Furthermore, the introduction to the field of artificial intelligence (AI) tools has been promoting a range of technological advances that keep on affecting the diagnostic potential of CE.

## 1. Introduction

Capsule endoscopy (CE) entered clinical practice in the year 2000 [1] and has since gained a sizeable place in gastrointestinal (GI) endoscopy. Each year, many publications are devoted to CE, including numerous conference presentations, several national and international guidelines, technical reviews and expert consensus opinions [2,3]. Manufacturers are steadily increasing technological specifications and diversifying CE models. Although small bowel (SB) CE remains the workhorse of CE [4], colon CE appears to get a firm grip in current years, and other models for the examination of the stomach or the stomach and the SB together, as well as those that provide panenteric visualization in one setting [5], have emerged.

In the currently available bibliometric studies, a surge in publications is noticed in many areas of medicine [6,7]. Furthermore, an inevitable shift in research topics and publication trends exists in the field of CE as well; while some of these topics attract steady interest, others may be just ephemeral or recurring trends. Therefore, retaining a clear overview of specific issues and themes in the literature is advantageous in terms of research planning. The surge of computational power has prompted a technique termed “text-mining”. In text-mining, computer algorithms extract text information: such methods can investigate trends in a research field [6]. This powerful technique allows for detailed and thorough data extraction superior to previously possible traditional bibliometric methodologies. Our study aimed to delineate the main trends in published CE research, focusing on shifts in topics, indications, and geographic developments—to our knowledge, this is the first text-mining analysis in the CE field.

## 2. Materials and Methods

Our overall approach is based on data-mining terms from scientific journals. Besides general data processing tools and exploration packages (R/RStudio v1.4.1717. https://www.rstudio.com/products/rstudio/release-notes/rstudio-1-4-juliet-rose/ (accessed date 17 July 2021)), we used the VOSviewer software (VOSviewer v1.6.18. https://www.vosviewer.com/ (accessed date 6 January 2022)), developed for scientific literature analysis (www.vosviewer.com) [8,9].

A search was performed in the Web of Science Core Collections (WoSCC) database with the keywords “capsule AND endoscopy” to create a relevant subset regarding the research area of interest. Out of the returned content of 7565 documents, only English-language papers, meeting abstracts, and proceedings papers were selected, with a remaining total of 5764 publications for the time span from 1 January 2000 to 31 December 2021. For all 5764 documents, a full bibliometric record, including cited references (58 field tags), was downloaded on 4 April 2022 (Figure 1).

### Data Preprocessing

First, we cleaned our dataset from records with missing publication years (*n* = 12) and documents published in 2022 (*n* = 18). Exploratory data analysis showed that only 48% of the documents had author defined keywords (field tag DE) and only 60% keywords plus (ID). A total of 2223 documents had no full-text, and abstracts and titles varied significantly in length (abstracts = 20–6242 characters; titles = 11–291 characters). Considering these limitations we decided to use both, keywords as well as abstracts and titles as units of analysis.

In preparation for the term analysis based on abstracts and titles, the dataset was split into 4 subsets for the time frames from 2000–2005, 2006–2010, 2011–2015, and 2016–2021. Both, the term analysis, observing occurrences of keywords and performed on abstracts and titles were supported with a gradually generated thesaurus file based on our findings in five consecutive preprocessing rounds. The thesaurus file served to omit general terms, disambiguate different spellings of identical terms, and cluster semantically related terms under meta labels.

## 3. Results

CE scientific literature has seen an overall annual percentage growth rate of 19.2%, with an average number of citations per document of 18.3, and a standard deviation of 80.6 publications per year over the time period of interest (Figure 2).

The largest share of the 5734 analyzed documents was published with regards to six research areas: gastroenterology and hepatology (*n* = 1954), surgery (*n* = 468), engineering (*n* = 463), radiology, nuclear medicine and medical imaging (*n* = 222), general and internal medicine (*n* = 223), computer science (*n* = 190), and pediatrics (*n* = 80).

A share of 54% of the articles, meeting abstracts, and proceedings were published by the top 10 sources with Gastrointestinal Endoscopy being the leading journal by far (32.8%) followed by the American Journal of Gastroenterology (15.4%) and Gastroenterology (10.8%). Table 1 provides an overview of the contributions by the 10 most productive authors: the table shows their ranking by Dominance Factor [10] as well as by the number of articles they contributed individually/respectively among the total 17,073 authors in the field of CE from 2000 to 2021.

The distribution over time for publications with reference to a selection of conditions/diseases is shown in Figure 3: Bleeding (*n* = 1105), Cancer and Tumors (*n* = 455), Celiac Disease (*n* = 132), Inflammatory Bowel Disease (IBD, *n* = 838), and Polyposis (*n* = 148) in our dataset of 5734 documents.

### 3.1. Geography

Our dataset includes contributions from 64 different countries measured by the first authors’ countries of origin with 1385 documents not specifying a first author and/or a first author’s country of origin and from 83 countries measured by the corresponding authors. According to the count based on first authors, most publications originated in the United States of America (*n* = 1048), followed by Japan (*n* = 592), China (*n* = 488), Italy (*n* = 309), United Kingdom (*n* = 234), Korea (*n* = 178), Germany (*n* = 176), France (*n* = 151), Spain (*n* = 117), and Israel (*n*= 112). Corresponding authors top 10 countries of origin included: USA (*n* = 1229), Japan (*n* = 585), United Kingdom (*n* = 461), Italy (*n* = 437), China (*n* = 436), Germany (*n* = 230), Korea (*n* = 186), France (*n* = 177), Israel (*n* = 173), and Spain (*n* = 171), Figure 4.

In extending our general exploration of bibliometric characteristics of CE publications over the past 20 years, we additionally performed a country collaboration network analysis (method: fractional counting) on our dataset of 5734 documents.

Figure 5 displays the standard normal distribution of publications over time for those countries, which contributed with at least 5 publications in the time of reference. The method fractional count splits the weight assigned for every multi-country-authored publication in fractions of one per collaborating partner. The node size represents the number of document weights per country, whereas the lines represent a collaboration between the countries they connect. The node color mirrors the average year of publication per country.

### 3.2. Topic Modeling

We performed a co-occurrence analysis based on the Web of Science meta-tags keyword plus and author-defined keywords, combined as a unit of analysis of all keywords in VOSviewer on our entire dataset of 5734 documents. KeyWords Plus are n-grams that frequently appear in the titles of the references cited in an article. KeyWords Plus are extracted from documents using an algorithm that is unique to Clarivate Analytics and is meant to provide an unbiased matching of keywords for articles [11]. The key term co-occurrence map in Figure 2 shows the frequency of the identified terms (node size) overlayed by coloring representing the standard normal distribution of the average publication year of the documents in which the terms appeared, respectively. The edge length connecting two nodes is proportional to the number of times two terms are found in proximity. The average publication year of the publications in which the term occurs is colored in a way that yellow indicates terms that occur mainly in recent publications and blue indicates terms that occur mainly in older publications.

The co-occurrence analysis (method: full counting) in VOSviewer initially returned 7957 keywords. These were listed in descending order of frequency and labeled as either specific or non-specific to CE research based on our thesaurus file. A total of 84 terms with a minimum occurrence of 15 times remained after the exclusion of general terms, disambiguation of spelling variations, and after grouping, terms under shared meta-labels/semantic units. In Figure 6, the distribution over the years of publications including these keywords is put on display. For visualizing the evolution of keywords in publications over time, we extracted the keyword-specific documents based on the WoSCC field tags author keywords and keywords plus from our core dataset.

With a second analysis (see Appendix A)—in form of a grouped display of the bibliographic data into four subsets for the periods from 2000–2005, 2006–2010, 2011–2015, and 2016–2021—we provide an alternative approach to showcase the topic’s evolution over time.

### 3.3. Capsule Types

Our final analysis takes us to the question of the variety of capsule types referenced in CE publications from 2000 to 2021 (Figure 7). White light imaging (WLI) has been the dominant diagnostic modality in CE so far. Over the last years there has been an increase in the application of non-WLI diagnostic imaging and sensing technologies to CE, since WLI technology is limiting the diagnosis to the mucosal surface of the gut. The integration of specific diagnostic imaging and non-imaging technologies (ultrasound, measurement of luminal pH, temperature, gas sensing and pressure) into capsule endoscopy devices enables, e.g., submucosal imaging and an improved differentiation between malignant and benign tissue [12].

## 4. Discussion

This study aimed to provide a general overview of the evolution of capsule endoscopy research based on bibliographic data since its beginnings over 20 years ago. Apart from gathering insights regarding the general distributions of publications and citations, we looked into general bibliometric characteristics. In addition, text-mining methods were used to explore the most frequent terms, capsule types and indications. All sub-analyses are available as Appendix A.

The overall number of publications had an annual growth rate of 19.2%, showing a continual rise until 2010 with a subsequent plateau. Regarding indications, we found a constant increase of publications related to bleeding, IBD and cancer/tumors, whereas celiac disease and polyposis marked a lowering (despite slow) trend throughout the years. This might be explained by the evolving approach in specific topics (e.g., panenteric CE in Crohn’s disease) [5] as well as the growing implementation of staging scores and AI tools for CE [13,14], also justifying the very recent upward shift seen in bleeding and IBD.

The analysis of the geographical distribution of publications showed some interesting points: countries like Japan and Italy have a high number of published papers (listing 2nd and 4th respectively in the overall list), despite their population is much lower than the USA and China (1st and 3rd); the same goes for Israel, that although being 10th it has a population fewer than 10 million people (as of 2021). These data are corroborated by the collaboration map (Figure 5), highlighting that the most productive countries are also those which collaborate the most in multicenter studies. Further benefits of international cooperation could be based on sharing of data—as described in the The FAIR Guiding Principles for scientific data management [15]. As shown in France, in the form of CAD-CAP (Computer-Assisted Diagnosis for CAPsule Endoscopy), data management is facilitated if centers collaborate which might also give an inspirational kick to other teams and/or manufacturers [16].

The analysis of topics (based on keywords) also provided interesting insights. The game-changer in CE over the last decade has been the introduction of increasingly advanced AI, as witnessed by the steady rise in Figure 6. Deep learning and convolutional neural networks can help to optimize the weaknesses of CE, which is the lengthy and tiring reading of lots of images, its harmonized classification and also localization [17,18].

Regarding the distribution of capsule type, it is interesting to notice how the number of studies on non-imaging capsules and robotic capsules has constantly been growing in recent years, whereas other types (i.e., esophageal capsule) marked a slow decline, thus showing a progressive loss of interest by the scientific community.

The visualization of the topical evolution in four-time segments (see Appendix A) gives a more detailed account of the development. On a meta-level, it shows the typical process of evolution in a field of scientific knowledge: from a period of forming (2000–2005), to a time of consolidation in numbers of publications and hence recognition (2006–2010), to a period of stronger diversification of contributions (2011–2017). What it cannot begin to show is the impact that advances in knowledge and understanding keep on having on the field. For example, immune-related, neural, and neuroendocrine communication channels support a correlational association between diet, gut, microbial dysbiosis, and various disorders such as depression [19,20] and Alzheimer’s [21].

CE’s role in prevention and in new therapeutic approaches for these other widespread diseases is yet to be determined [22], though first steps showing the possibility to collect gut bioinformation with a magnetically controlled sampling capsule are already complete [23]. Even though most of the newer capsule types require further testing, we expect that these and potentially further experimental capsule designs will open up new routes for e.g., improved computer-aided diagnosis and virtual biopsy/sampling, enhanced capsule localization with clear benefits for future clinical practice, and as a technology for research towards a better understanding and treatment of complex/systemic diseases/conditions, e.g., like those involving the circuitries of the gut-brain axis [24,25].

Further developments in the clinical application of CE concern, e.g., new routes of supply, simplified patient pathways and an increasing focus on a diverse patient population. The FDA granted at-home administration amidst the global COVID-19 pandemic with already significant impact in endoscopy departments to delivery [26,27]. And while in China the capsule endoscopy group of the Society of Digestive Endoscopy published a recommendation on indications for urgent, semi-urgent and elective capsule endoscopy examination during the COVID-19 pandemic [28], Scotland was the first to implement a national rollout service in colon capsule endoscopy (CCE) [29]. CE has proven to be beneficial as a minimal-invasive, relatively safe approach with high diagnostic yield for vulnerable patient communities, e.g., pediatric populations [30], and it can be hoped for that benefits of outpatient solutions in combination with advances in telemedicine will translate to further, so far hard to reach populations worldwide [31].

Regarding limitations of our study, it remains to be added that the relationship between the literature and the community producing it is far more complex than what text-mining methods can offer. The number of documents, and citations depends on the bibliometric database, and the analysis tools used. Research and development in areas with high societal impact might benefit from the inclusion of data from a more diverse range of sources. Our study shows several limitations: (i) analyses with VOSviewer only include articles in English language, thus excluding from the study a -possibly- consistent amount of studies; (ii) as the article indexing provides data on the first authors only, it is not possible to base the analysis on the corresponding ones which are often the better representatives of the articles’ country of origin; (iii) data-mining approach is neither, completely objective nor subjective, instead: interpretive and data-driven. Also, keywords provided by the authors are not standardized and sometimes not clearly classifiable, e.g., what is meant with “pathology” or “disease”, or mentioning “wireless capsule” is not enough without knowing the capsule type such as imaging versus non-imaging capsule.

In conclusion, our analysis confirms that the SB capsule remains the workhorse of the capsule endoscopy field [32]. Developed in the mid-1990s and approved by FDA for clinical use in 2001, it is now the first-line diagnostic tool for the investigation of SB diseases [2]. The colon capsule (FDA-approved in 2014), intended to be used for detection of colon polyps in patients after incomplete optical colonoscopy, has gained in importance over the years, while research on other devices (e.g., the esophageal capsule) seems to have reached maturity and/or has lost relative importance in the face of a range of more recent developments such as the Robotic Capsule (RC), Magnetic Capsule (MCE) (as shown in Figure 7). Despite its methodological restrictions, the text-mining techniques and tools applied in our review offer a fruitful approach for exploring the dynamics in the field of CE throughout the last two decades showcasing the impact of AI and progress from diagnosis to therapy [33].

## Figures and Tables

**Figure 1 diagnostics-12-02238-f001:**
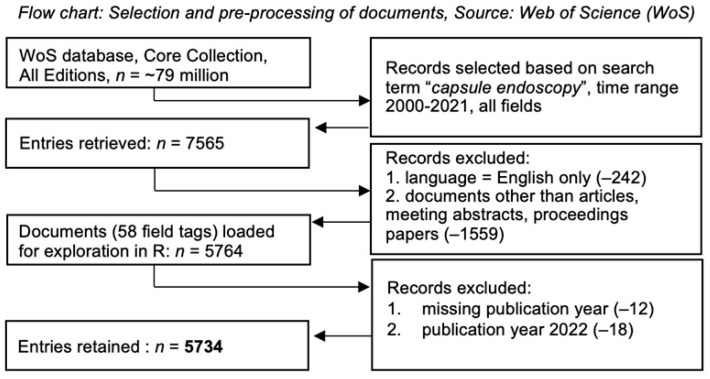
Flowchart of the study.

**Figure 2 diagnostics-12-02238-f002:**
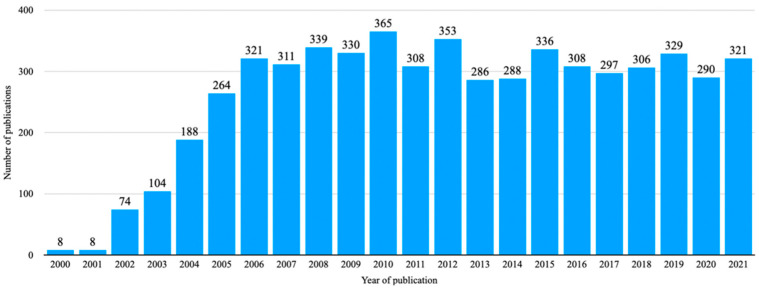
Capsule endoscopy publications, 2000 to 2021.

**Figure 3 diagnostics-12-02238-f003:**
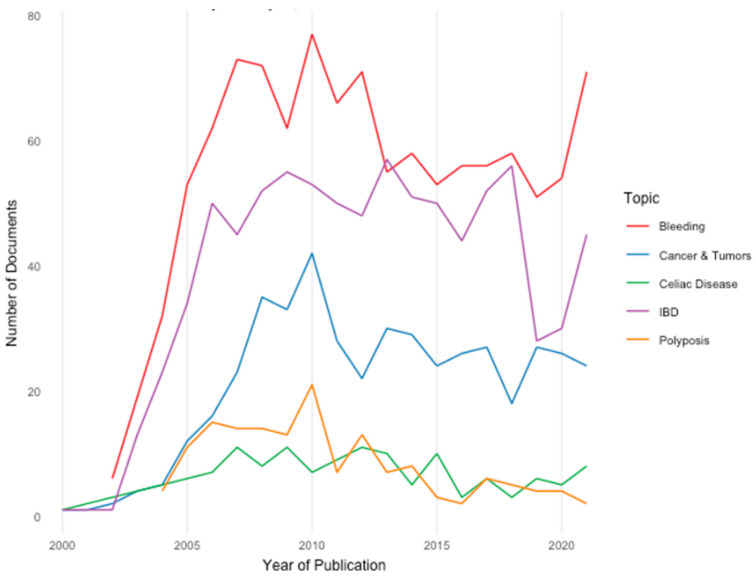
Capsule endoscopy publications per topic, 2000 to 2021.

**Figure 4 diagnostics-12-02238-f004:**
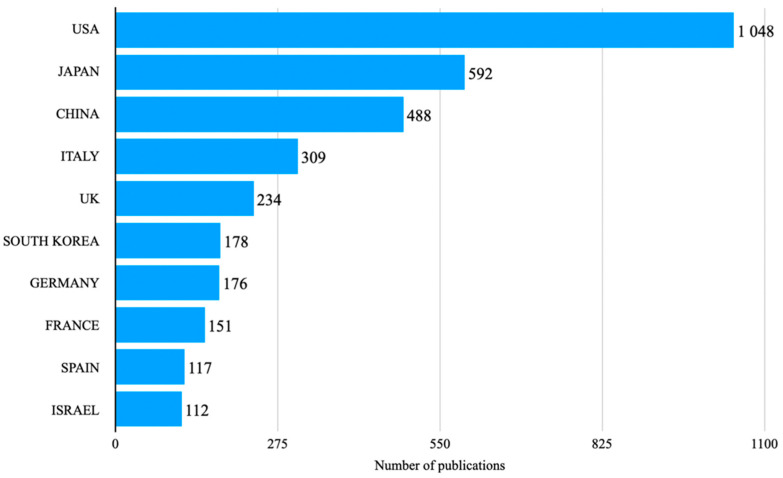
10 most productive countries in CE publications, 2000 to 2021.

**Figure 5 diagnostics-12-02238-f005:**
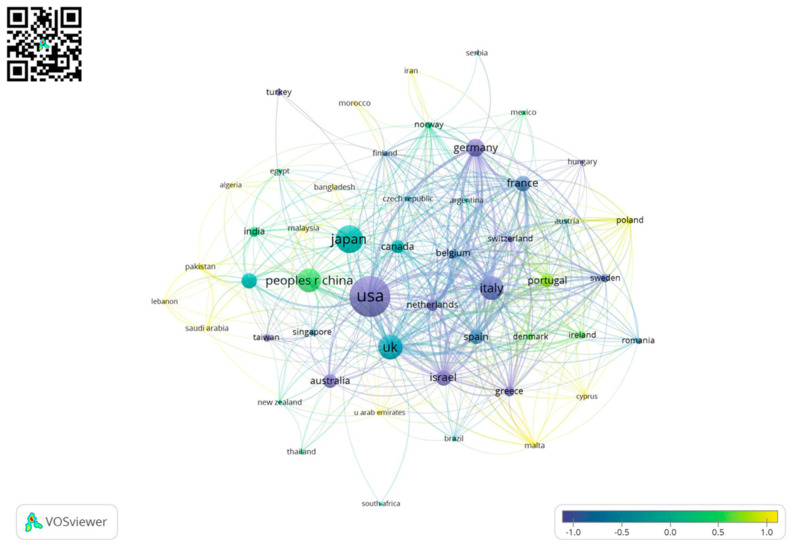
Country collaboration map. **Note**: Layout settings VOSviewer: Attraction = 2; Repulsion = −2; Resolution = 1; minimal Cluster size = 1. Legend shows standard normal distribution of average publication year per country. The visualization is available online at https://tinyurl.com/y6ofc5jd.

**Figure 6 diagnostics-12-02238-f006:**
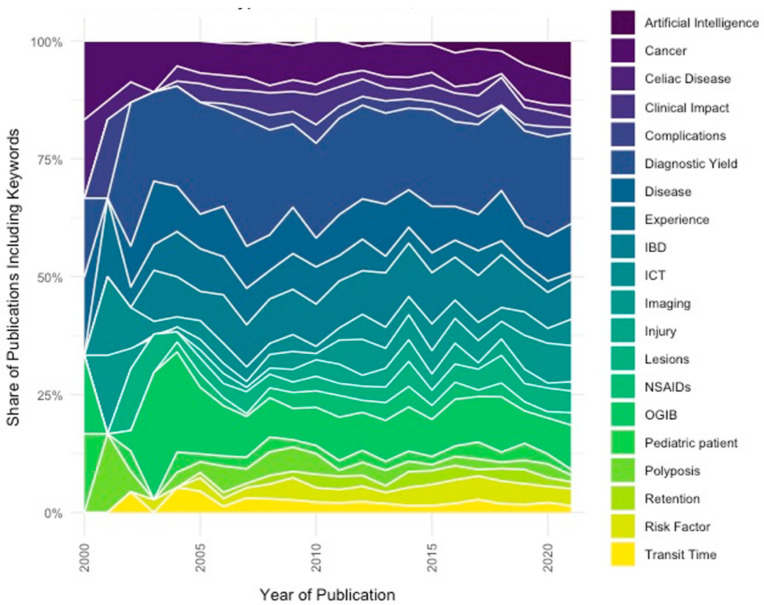
Capsule endoscopy publications per keywords, 2000 to 2021.

**Figure 7 diagnostics-12-02238-f007:**
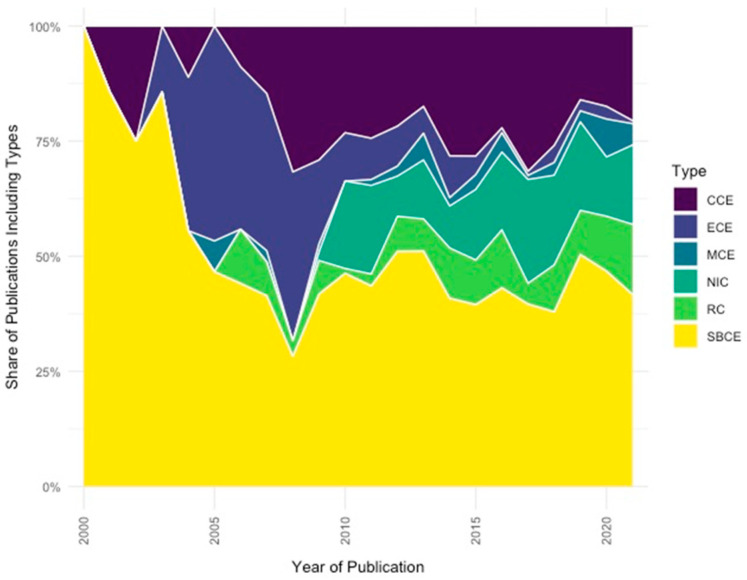
Capsule endoscopy publications per type of capsule, 2000 to 2021. **Note**: Robotic Capsule (RC), Magnetic Capsule (MCE), Colon Capsule (CCE), Esophagus Capsule (ECE), Non-Imaging Capsule (NIC), Small-Bowel Capsule (SBCE).

**Table 1 diagnostics-12-02238-t001:** 10 most productive authors in the field of CE, 2000 to 2021.

Author	Dominance Factor	TAA	SAA	MAA	FAA
Spada, C.	0.47	79	0	79	37
Nakamura, M.	0.38	72	0	72	28
Sidhu, R.	0.31	84	0	81	25
Koulaouzidis, A.	0.28	68	0	68	18
MC Alindon, M.E.	0.23	79	0	79	0
Leighton, J.A.	0.22	61	2	59	17
Eliakim, R.	0.17	71	2	69	16
Costamagna, G.	0.07	62	0	62	5
Cotter, J.	0.04	61	0	61	2
Tanaka, S.	0.014	73	0	73	1

**Note**: Dominance Factor = Number of First Authored Articles (FAA)/Number of Multi-Authored Articles (MAA); Multi-Authored (MAA) = Total Number of Articles (TAA) − Number of Single Authored Articles (SAA).

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
