# Peer review of "An Overview of the Evolution of Capsule Endoscopy Research—Text-Mining Analysis and Publication Trends"

_diagnostics, 2022, doi:10.3390/diagnostics12092238_

Round 1

Reviewer 1 Report

In this article, Steinmann et al. performed a bibliometric analysis of the evolution of CE publications during the past two decades. The researchers identified over 5000 articles and highlighted an increasing research effort in the field. The study is methodologically sound and well written. All appropriate sub-analyses were performed, while the provided figures and tables enhanced the readability of the manuscript. However, prior to publication, the authors should consider the following issues:

- Please remove all text highlighted in red

-Please provide a subgroup analysis regarding the type of research (experimental, clinical trial, review/meta-analyses). This will provide significant information of the research direction during the analysis period and increase the scientific value of the manuscript.

Reviewer 2 Report

  • The English language of the article needs to be revised. Namely, there are some unclear sentences, and also there are a few instances of using phrases that are not typical for academic medical writing.
  • Lines 25-28: Could you clarify this in the context of your study? Both sentences, especially the second, seem like they need to be put in the context and seem unfinished. Revise - just add more specific details - for example, the second sentence is not clear enough - specify that the importance of AI refers to the CE application.
  • Introduction: This section needs to be expanded. The authors do not provide a comprehensive overview of the field. The fact that only three references are cited throughout the entire Introduction confirm this.
  • Lines 45-46: This sentence is too broad - be specific, provide references, mention some fields. More importantly, have there been any previous bibliometric studies on CE? If not, this underlines the novelty of your study - and this too should be mentioned here.
  • Line 47: Why "even"? Why would it be surprising that such shifts exist in this field compared to other fields?
  • Line 49: Why is it advantageous and what for? Explain.
  • Lines 63-64: Why were reviews not of interest to include in the literature search?
  • Line 65: Specify the inclusion timeframe further - e.g. is it Jan 1st 2000 - Dec 31st 2020? Or actually end of 2021?
  • Figure 1: Clarify within the cells which explain which records were excluded - that you actually excluded documents that were NOT articles, meeting abstracts or proceedings papers.
  • Line 74: How did you count meeting abstracts - as articles having abstracts or not? Clarify.
  • Methods section needs to provide a little more detail on, e.g., literature characteristics which were explored?
  • Lines 93-96: The sentences here seem as if they are misplaced. They do not represent results, they seem more like they belong to the section Discussion. In addition, references are missing here.
  • Line 101: The words and brackets in this sentence seem misplaced - check and revise.
  • Table 1: Make the Table 1's title more specific and informative - add that these are authors of what - research publications on CE.
  • Figure 4: Make the title of this figure more informative - specify what the countries are most productive in - articles on CE.
  • Lines 188-202: This text represents your explanations of your findings and not the findings themselves, so please move this to the Discussion.
  • Lines 282-285: The conclusion is too general - it needs to be based on your findings.

Round 2

Reviewer 2 Report

I would like to thank the authors for addressing my comments. They have improved the sections Introduction and Discussion, edited Results. In addition, in an attempt to address comments they've performed additional analyses. I agree with the authors regarding the subgroup analysis. 

  Minor comments: Conclusion - please reconsider the use of references in this section. In addition - consider whether it would be appropriate to omit the part regarding research on other devices reaching maturity (if it's not a finding of the present paper).
